# Quantification of flexoelectricity in PbTiO$_3$/SrTiO$_3$ superlattice polar vortices using machine learning and phase-field modeling

Q. Li [1], C.T. Nelson[2,3], S.-L. Hsu[2], A.R. Damodaran [2], L.-L. Li[1], A.K. Yadav [2], M. McCarter[4], L.W. Martin [2,5], R. Ramesh[2] & S.V. Kalinin [1]

Flexoelectricity refers to electric polarization generated by heterogeneous mechanical strains, namely strain gradients, in materials of arbitrary crystal symmetries. Despite more than 50 years of work on this effect, an accurate identification of its coupling strength remains an experimental challenge for most materials, which impedes its wide recognition. Here, we show the presence of flexoelectricity in the recently discovered polar vortices in PbTiO$_3$/SrTiO$_3$ superlattices based on a combination of machine-learning analysis of the atomic-scale electron microscopy imaging data and phenomenological phase-field modeling. By scrutinizing the influence of flexocoupling on the global vortex structure, we match theory and experiment using computer vision methodologies to determine the flexoelectric coefficients for PbTiO$_3$ and SrTiO$_3$. Our findings highlight the inherent, nontrivial role of flexoelectricity in the generation of emergent complex polarization morphologies and demonstrate a viable approach to delineating this effect, conducive to the deeper exploration of both topics.

[1] Oak Ridge National Laboratory, Institute for Functional Imaging of Materials and Center for Nanophase Materials Science, Oak Ridge, TN 37831, USA. [2] Department of Materials Science and Engineering, University of California, Berkeley, CA 94720, USA. [3] Materials Science and Technology Division, Oak Ridge National Laboratory, Oak Ridge, TN 37831, USA. [4] Department of Physics, University of California, Berkeley, CA 94720, USA. [5] Materials Sciences Division, Lawrence Berkeley National Laboratory, Berkeley, CA 94720, USA. Correspondence and requests for materials should be addressed to Q.L. (email: qianli.ornl@gmail.com) or to S.V.K. (email: sergei2@ornl.gov)

Flexoelectricity is an early-documented polarization mechanism that accounts for the coupling effect between heterogeneous material strains and electrical polarization[1–5]. In contrast to ferroelectricity which is known to only exist in materials with noncentrosymmetric crystal structures, flexo-electric polarization can develop in all materials including, for example, canonically nonpolar $SrTiO_3$ bulk crystals[6]. This is because the presence of strain gradients essentially breaks peri-odic lattice translational symmetries, thereby leading to uncom-pensated bound charge across multiple (undeformed) unit cells. By virtue of its ubiquity, flexoelectricity holds considerable implications for designing polar materials as utilized in many electromechanical devices, especially in the forms of epitaxial thin films and other nanostructures[7]. In these forms, large strain gradients far exceeding those possibly accommodated in bulk phases can arise and strongly enhance the flexocoupling strength at both nano- and mesoscales[8,9].

Theoretically, a systematic range of structural and dynamical behaviors of materials can be modulated by flexoelectricity, including phonon softening and associated structural instabilities[10–13], polarization switching[14–16], domain wall for-mation[17,18], and as charge carrier transport[19]. Experimental observations based on scanning probe, scattering, and bulk

measurement techniques have provided compelling evidence of these flexocoupling effects[6,11–13,19]. However, those studies typi-cally exhibit large uncertainties and, so far, flexoelectric coeffi-cients have only been reliably measured for a limited number of materials. This deficiency is further complicated by technical difficulties in separating flexoelectricity from other intimately intertwined polarization effects, such as interface contributions and chemical heterogeneities[20,21]. As opposed to direct electro-mechanical characterizations, examinations of local structural distortions have been shown to be an effective method for eval-uating the flexocoupling strengths in a few materials[9,22].

Very recently, an ordered vortex-array topology of electric polarization was observed in $PbTiO_3$/$SrTiO_3$ (PTO/STO) thin-film superlattices, which has significantly extended our current view of complex topological states in polar materials and potentially leads to emergent functionalities[23]. This vortex topology primarily features the polarization vector rotating con-tinuously across several nanometers, rather than being con-strained along the <001> as would be presumed for the prototypical tetragonal ferroelectric PTO. Over a longer length scale, these vortices occur in a large population and exhibit mesoscopic correlation and ordering to an extent rarely observed for conventional ferroelectric domain assemblies in other

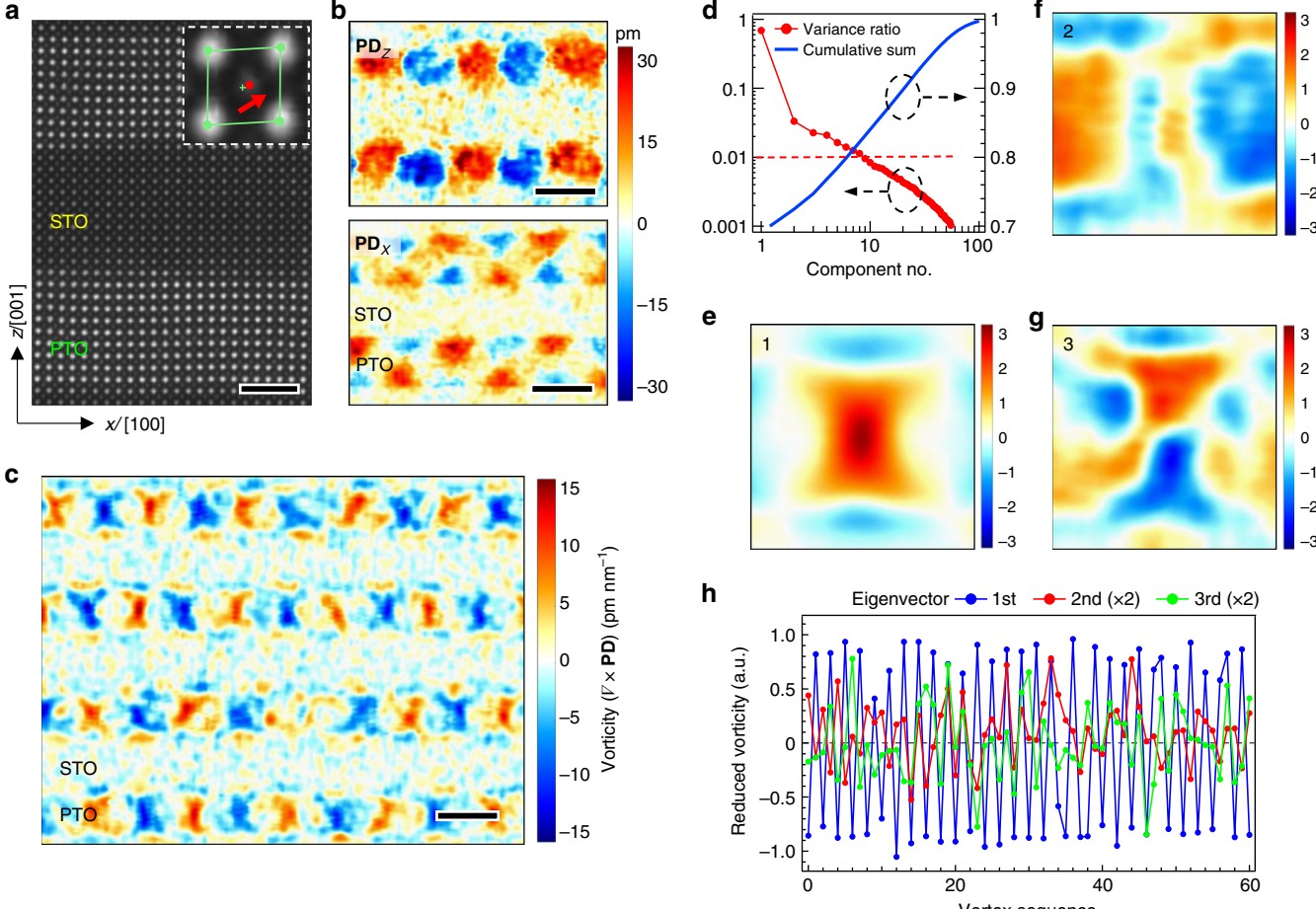

**Fig. 1** Multivariate analysis of the observed polar vortex patterns. **a** Typical atomic-resolution high-angle annular dark-field (HAADF) STEM image of $(PTO)_{10}$/$(STO)_{10}$ superlattices. Inset is a zoom-in view of a single unit cell with the Pb/Ti atom positions denoted by green/red dots and the center-of-mass position of the Pb sublattice by a green cross; from their relative shift, the polar displacement (**PD**) vector can be derived as denoted by a red arrow. Scale bar, 2 nm. **b** The z/x-components of the **PD** vector field for a selected sample region. Scale bar: 5 nm. **c** Typical experimental vortex patterns represented by the vorticity calculated from the **PD** vector field. Scale bar, 5 nm. **d–h** Selected PCA results of the entire vortex dataset, of which **d** is the explained variance ratio for ~60 principal components, **e–g** are the first three components (loading map size: $5.4 \times 5.4$ nm$^2$) and the corresponding eigenvectors are shown in **h** with the ordinates rescaled

epitaxial thin films[9,24]. Therefore, the PTO/STO polar vortex constitutes a unique system for exploring the interactions between the order parameters and intrinsic material properties as well as extrinsic lattice disorder effects.

Here we elucidate the influence of flexoelectricity on the polar vortex formation process in PTO/STO superlattices based on phenomenological phase-field modeling informed by machine-learning analysis of atomically resolved structural imaging data[25]. Through quantitative matching of experiment and theory, we first reveal the occurrence of flexoelectricity-driven fine topological features hosted within the characteristic structural configuration of the PTO/STO polar vortices. Such a revelation, in a self-consistent manner, reflects the flexocoupling strengths of the constituent materials and allows for their quantifications. Our study thus demonstrate an approach to developing a comprehensive and predictive understanding of physical coupling phenomena for a potentially broad family of materials.

## Result

**Statistical analysis of polar vortices.** We begin with a brief description of the extraction of the polar vortices through atomic-resolution scanning transmission electron microscopy (STEM), as detailed in a previous report[23]. Fig. 1a shows a typical cross-sectional region of (001)-oriented $(PTO)_{10}/(STO)_{10}$ (the pseudocubic notation is used in this article; the subscript denotes the number of unit cells) superlattices epitaxially grown on $DyScO_3$ substrates, imaged along the [010] zone axis. The $A$-site Pb/Sr and $B$-site Ti sublattices are resolved by Z-contrast in the STEM images, and based on their mutual offset, a polar atomic displacement vector (**PD**; see Fig. 1 inset) can be derived for each unit cell as a faithful projection of its electric polarization $\mathbf{P}$[26]. Fig. 1b illustrates the $\mathbf{PD}_Z$ and $\mathbf{PD}_X$ components featuring nearly round and triangle-shaped, respectively, ordered pair patterns with alternating polarities within the PTO layers. The **PD** of the STO layers shows no obvious patterns, but at least, there are random finite-magnitude features indicating partial penetration of the ferroelectric distortion into the STO layers. To quantify the vortex structure, we further introduce vorticity, namely the curl vector of the **PD** in the $xz$-plane, $(\nabla \times \mathbf{PD})_{[010]} = \mathbf{PD}_{z,x} - \mathbf{PD}_{x,z}$ (see Methods for details). Fig. 1c shows the calculated vorticity over a region containing some 40 vortices. Overall, these vortices follow a strict in-plane long-range order; namely, the vortex pairs exhibit a periodicity of 9–10 nm. By contrast, their out-of-plane alignment across multiple PTO layers appears to be much less strict; for example, here the top two vortex layers are in anti-phase with the bottom two. Apart from this weak interlayer coherence, most vortices are also observed to have nonideal shapes with certain extent of spatial variations, precluding a simple unified description of their individual topological characteristics.

To extract statistically significant information from the observed polar vortices, we analyze their patterns collectively using principal component analysis (PCA). As an unsupervised learning procedure, PCA decomposes $N$ possibly correlated observations into a set of linearly uncorrelated, orthogonal eigenvectors $w_j$ (or principal components; $j = 1,...,N$) as described by equations $N_i = a_{ij}w_j$, where the expansion coefficients $a_{ij}$ are termed component loadings[27,28]. These eigenvectors are successively arranged according to the descending order of their eigenvalues. As such, the first PCA component contains the maximum variance (which equates to the majority content of information) and each subsequent component contains most of the remaining variance after subtraction of all preceding ones. Here in our PCA analysis, an observation corresponds to one single vortex patch segmented from the whole imaged sample

region (see Methods for the segmentation scheme), and in total a dataset of ~130 square patches were gathered from three STEM images taken on the same sample.

Figure 1d is the scree plot of the explained variance ratios of the major PCA components obtained from the above dataset, and the first three loading maps are illustrated in Fig. 1e–g. The first component explains ~70% of the variance and in its loading map, a vortex pattern is fully restored (Fig. 1e). This pattern bears the greatest similarity among all the observed vortices and hence best represents the global characteristics of the system. The variance drops sharply to ~3% for the second component and afterwards decays very slowly. Those corresponding PCA components reflect the vortex shape variations and may pertain to several aspects or factors. A somewhat systematic aspect is the nonuniform periodicity (thus the vortices have slightly different aspect ratios) as seen in both STEM and diffraction measurements, which largely accounts for the misalignment features in the second/third (as well as the fourth) loading maps primarily anti-symmetric about the [100]/[001], respectively (Fig. 1f, g). All the after-fourth loading maps are dominated by randomly scattered, spot-like features (see Supplementary Fig. 1 for the 4–16 components). Their corresponding variances are around or below 1% but altogether amount to ~21%, which may well overweigh noises from the imaging and data analysis processes and suggest the contributions of highly spatially localized factors, such as random lattice defects within the imaged atomic columns. Moreover, the spatial distributions of these component features over multiple vortex regions are indicated by their respective PCA eigenvectors. As depicted (in portion) in Fig. 1h, the first eigenvector shows regular oscillations clearly associated with vortex/anti-vortex pairs, while the second and third ones show less straightforward modes of variation. Note that all these spatial distribution modes have no cross-correlations as determined by PCA.

**Flexoelectricity-coupled phase-field modeling.** To gain mechanistic insights into various global and local behavior of the PTO/STO superlattices identified from the multivariate analysis of the STEM results, we have performed phase-field modeling within the Landau–Ginzburg–Devonshire (LGD) theory framework (see model details in Methods)[29–31]. Here, we concentrate on the influence of flexoelectricity which by definition is an intrinsic material property acting globally on the system, unconsidered in previous studies[23,32]. Per their cubic parent phase symmetries, the flexoelectricity of both PTO and STO is comprised of three components: the longitudinal, transverse, and shear coupling with coefficients $f_{11}$, $f_{12}$, and $f_{44}$, respectively. In the following, we explore the three types of flexocoupling in the PTO and STO layers separately by varying their coefficients within a tentative range and in doing so, compare their apparent modulation effects to the vortex structure.

Figure 2a presents a portion of the simulated vorticity, $(\nabla \times \boldsymbol{P})_{[010]} = \mathbf{P}_{z,x} - \mathbf{P}_{x,z}$, of the $(PTO)_{10}/(STO)_{10}$ superlattice in the absence of flexoelectricity (i.e., all $f$'s equal zero). A full-view animation showing the dynamic evolution of vortices from random initial values is presented in Supplementary Movie 1. The vortex layers in this region are highly ordered, only with the presence of an anti-phase boundary. Still, the overall vortex arrangement of simulated models is found to be somewhat dependent on the initial values, and occasionally exhibits a mixture with $a$-domains where the **P** vector lies purely in-plane for one or two PTO layers. These results agree well with a recent report by Hong et al.[32], pointing to a delicate free-energy landscape. In any case, our model captures all essential attributes of the observed vortex assembly, including the mixed periodicity (9–10 nm) and partial interlayer ordering state.

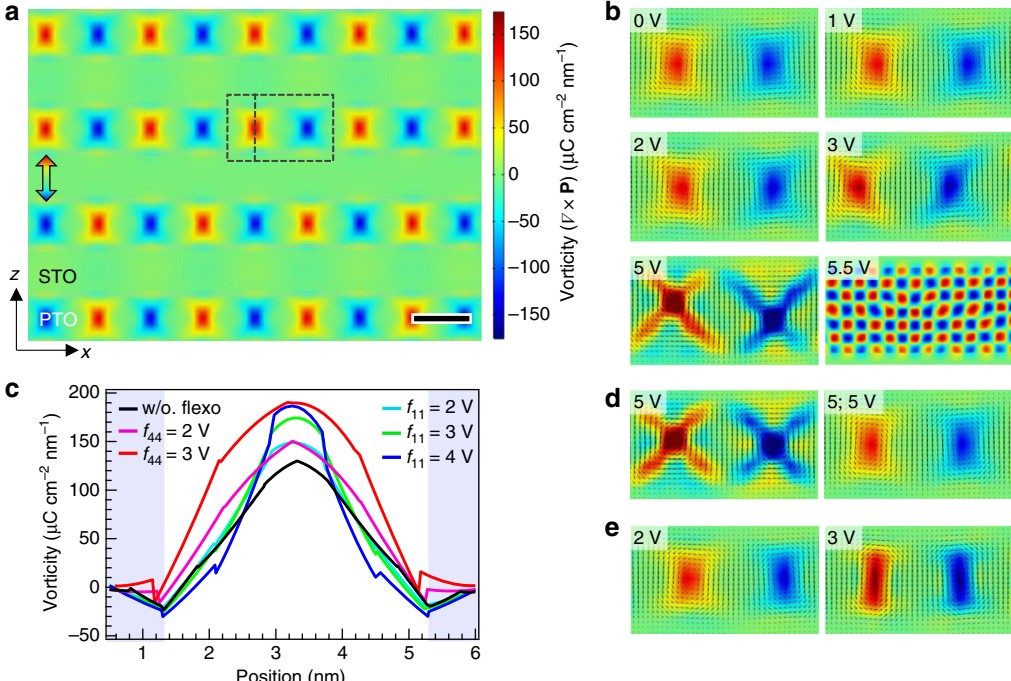

**Fig. 2** Modeling the flexocoupling effects of PbTiO₃. **a** Simulated vortex patterns represented by the vorticity calculated from the polarization vector field. A vortex/anti-vortex pair region marked by the dashed line is chosen to illustrate the flexcoupling effects in what follows. The double-arrow symbol indicates an anti-phase boundary. Scale bar, 5 nm. **b**, **d**, **e** The evolution of the vortex pattern under varying longitudinal **b**, transverse **d**, and shear **e** flexocoupling of the PTO layers. In **d**, "5; 5 V" denotes the case of $f_{11}^{PTO} = f_{12}^{PTO} = 5.5$ V; all others have only one nonzero $f$ variable. The same color scale is used here as that for **a**, except for $f_{11}^{PTO} = 5.5$ V in **b** whose range is from −1,000 to 1,000 ($\mu$C cm$^{-2}$ nm$^{-1}$). The polarization vectors (arrows) are overlaid on the vorticity images. **c** Section line profiles of the vorticity along the line marked on **a** (in-plane shift corrected), under selected flexocoupling conditions. The lightly blue-shaded region corresponds to STO regions

As the longitudinal flexocoupling of PTO ($f_{11}^{PTO}$) is increased from 0 to 5 V, the vortex shape gradually changes in such way that the vorticity becomes concentrated along the $xz$-plane diagonal directions, as shown in Fig. 2b for the same model region. Fig. 2c depicts the vorticity across the core of a single vortex along the [001]. The core vorticity increases gradually with $f_{11}^{PTO}$ until at 5 V, where it surges to about twice that for 0 V (~130 $\mu$C cm$^{-2}$ nm$^{-1}$) and meanwhile the vortex pairs show marked tilting and asymmetric distortion. A further increase of $f_{11}^{PTO}$ to 5.5 V results in fragmentation of the vortices and formation of a modulated structure with a fine checkerboard-like polarization pattern (Fig. 2b). The issue of flexoelectricty-induced modulated phases has been discussed at length in the literature[13,17]. Here its occurrence above certain critical $f$'s value can indicate the upper bounds of their realistic coefficients, since there are no such corresponding structures observed experimentally thus far. Besides the shape modulation, the coupling of $f_{11}^{PTO}$ also causes rigid in-plane shifts to the vortices, presumably through rebalancing of long-range electrostatic and elastic interactions in the system.

As a result of its unique structural configuration, the PTO/STO polar vortex is found to show interesting symmetry behavior in the flexocoupling. Fig. 2d illustrates the transverse flexocoupling for $f_{12}^{PTO} = 5$ V, exhibiting a distorted vortex shape in high resemblance with that for $f_{11}^{PTO} = 5$ V. When both coefficients are equally present (5 V shown here, for instance), the resultant vortex acquires a shape nearly identical to the zero flexocoupling case, indicating anti-symmetric effects of $f_{11}^{PTO}$ and $f_{12}^{PTO}$. This behavior is underlained by the fact that a continuous rotation of the **P** vector inside the vortices leads to zero polarization divergence (i.e., $\nabla \cdot \mathbf{P} = 0$) together with a highly symmetric strain and strain-gradient profile in the $xz$-plane. These two factors

effectively cancel out the flexoelectric energy contribution in the case of equal $f_{11}^{PTO}/f_{12}^{PTO}$ (see Eq. (2) in Methods). Such symmetry behavior implies that only a net effect of the longitudinal and transverse flexocoupling, ($f_{11}^{PTO} - f_{12}^{PTO}$), can be extracted from the experimental observations.

In contrast with the effects of $f_{11}^{PTO}/f_{12}^{PTO}$, the shear flexocoupling by $f_{44}^{PTO}$ essentially concentrates the vorticity along the out-of-plane direction leading to elongation of the vortex shape, as illustrated for $f_{44}^{PTO} = 2, 3$ V in Fig. 2e. Likewise, another modulated structure can be induced above a critical $f_{44}^{PTO}$ value of 3.5 V (not shown here) suggesting the practical upper bound of this coefficient. In addition, we have also compared the cases of negative $f$'s of PTO (as well as those of STO) since in principle there is no positivity constraint for flexoelectric coefficients[3]. These negative $f$'s are found to mainly change the sense of the vortex distortions, as expected, without introducing new distortion modes. Hence, this region of parameter space is not to be further explored.

We next examine the flexoelectricity of the other constituent material, STO. During the vortex formation process, the STO layers regulate elastic deformations of the PTO layers arising from an electrostrictive coupling of polarization in the latter; meanwhile, to satisfy polar continuity across the interfaces, STO can be strongly polarized by the PTO layers even at room temperature as facilitated by its incipient polar soft mode. Thereby both strain and polarization gradients exist in the STO layers, especially near the interfaces (see Supplementary Fig. 3 for the measured and simulated strain distribution maps). The presence of flexocoupling can adjust these gradients and underlying polarization and strain states, possibly leading to a two-way modification of the PTO layers. Our modeling shows that the longitudinal and transverse flexocoupling of STO ($f_{11}^{STO}/f_{12}^{STO}$) have minimal effects on the

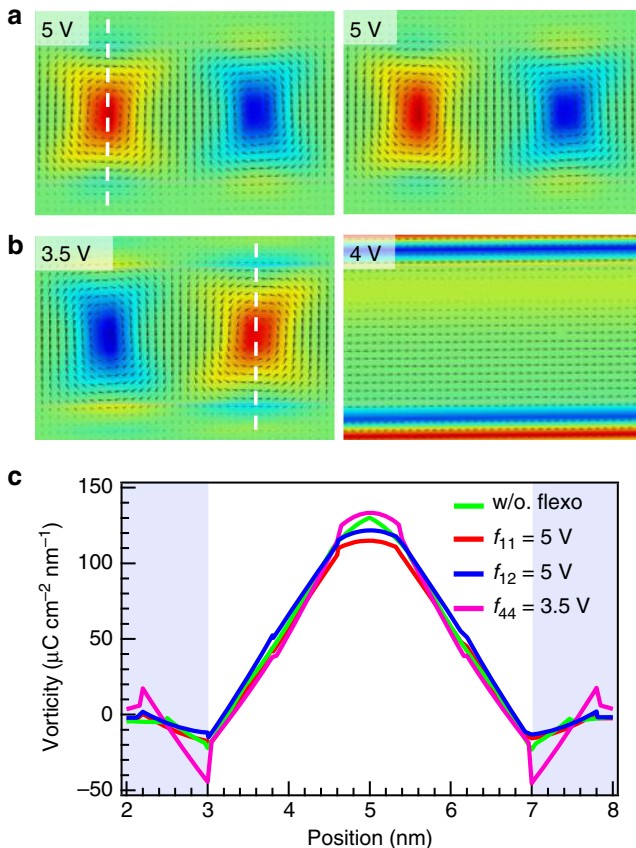

**Fig. 3** Modeling the flexocoupling effects of SrTiO$_3$. **a**, **b** Typical vortex pair pattern (location marked in Fig. 2a) under the flexocoupling of the STO layers: $f_{11}^{STO} = 5$ V (**a**, left), $f_{12}^{STO} = 5$ V (**a**, right), and $f_{44}^{STO} = 3.5, 4$ V (**b**). The polarization vectors (arrows) are overlaid on the vorticity images. **c** Section line profiles of the vorticity along the line marked in **a**, **b** under selected flexocoupling conditions. The lightly blue-shaded region corresponds to STO regions

vortex structure even for a high coefficient of 5 V, as illustrated in Fig. 3a, c; in both cases, the resultant vortex patterns show negligible difference from the 0 V case.

By contrast, the shear flexocoupling of STO has a more significant influence on the system. As shown in Fig. 3b, c, the vorticity near the interfaces is gradually enhanced with increasing $f_{44}^{STO}$, reaching to ~30% of the vortex core value at 3.5 V. In the PTO layers, this coupling also lowers the symmetry between the vortex pairs by changing their relative sizes and tilts. The $f_{44}^{STO}$ coefficient exceeds its critical value at 4 V, producing a layer-like modulated phase in the STO layers and simultaneously transforming the vortex state into an $a$-domain-like structure. Furthermore, a strong $f_{44}^{STO}$ coupling can promote the out-of-plane long-range ordering of the vortices via a substantial reconditioning of the STO layers. Supplementary Fig. 4 depicts that two anti-phase boundaries formed under zero flexocoupling are annihilated under $f_{44}^{STO} = 3.5$ V, and that all vortices seem to be "reconnected" via a regular, sheared network of STO polarization.

**Matching experiment and modeling.** The phase-field modeling reveals distinct vortex shape and long-range ordering modulation modes linked with different types of flexocoupling of PTO/STO. This allows for an evaluation of their coupling strengths by comparing the observed vortex patterns with simulated ones. In light of those results, we identify three flexocoefficients, $f_{11}^{PTO}$ (or $f_{12}^{PTO}$), $f_{44}^{PTO}$, and $f_{44}^{STO}$, that effectively control the vortex

structure. Note that the co-action of these coefficients is inherently nonorthogonal and nonlinear, although their individual shape characteristics appear to be preserved in a qualitative manner. Therefore, we have simulated a library of models enumerating all combinations within their established realistic value bounds. For quantitative comparison, we have analyzed the simulation results using the same procedure, PCA included, as for the experimental data. By doing so, we can factor in small nonuniformity in the simulated vortex assemblies (which mostly have >98% variance for the first component) and enhance their spatial resolution practically limited by the mesh density of our model. We further adopt the structure similarity (SSIM) index for measuring the similarity between two images, which ranges from −1 to 1 with increasing similarity (see details in Methods)[33].

Fig. 4a, c provides a visual comparison between the observed and simulated vortices, represented by their first PCA components. A large mismatch along the [100] can be found for the model with zero flexocoupling, evidencing the presence of this effect. Indeed, the calculated SSIM index shows systematic improvement as $f_{11}^{PTO}$ and $f_{44}^{PTO}$ approach the global best-fit values and drops quickly after they overshoot (Fig. 4b). The dependence of this index on $f_{44}^{STO}$ appears to be not so well defined here, due to exclusion of the STO region in the SSIM calculation. We have exercised this exclusion because the STEM image-derived **PD** vector could have undervalued the STO polarization, relative to that of PTO (the former has larger dependence on the oxygen sublattice arrangement), and thus a rigorous linearity may not hold between them. Nonetheless, this factor can be improved by further imaging studies with full detectability of STO polar distortions. Here an additional rationale for evaluating $f_{44}^{STO}$ is its influence on the global vortex ordering state; the weaker interlayer coherence observed from STEM suggests its value being not as high as 3.5 V (cf. Fig. 1c and Supplementary Fig. 4).

## Discussion

The flexoelectric coefficients obtained from the vortex shape analysis, $(f_{11} - f_{12}) \sim 3$ V and $f_{44} \sim 2$ V for PTO and $f_{44} \leq 3$ V for STO, are in good agreement with several recent theoretical studies[3,34–36], for example, first-principle calculated $|f_{11} - f_{12}| = 1.24$ V and $|f_{44}| = 1.96$ V for STO[35]. These values also reinforce an early prediction made by Kogan that $f$ should be on the order of $q/a$ (~1–10 V, where $q$ is electron charge and $a$ interatomic spacing)[1,5], and may serve as a benchmark for future studies. Note that our analysis builds upon global statistics of the observed vortex assembly and invariably has less uncertainty than those estimated from single, isolated observations of structure distortion[22]. Furthermore, compared to regular ferroelectric domain walls[9,17], the polar vortex topology appears to be much more sensitive to the presence of flexoelectricity since it inherently hosts a set of large, coordinated strain and polarization gradients and begets strong mesoscale elastic and electrostatic correlations. Such delicate energetics renders the system susceptible to the flexocoupling, primarily via renormalization of the polarization gradient energy (which directly links with direction and/or magnitude changes of the polarization vector). All in all, flexoelectricity plays an integral part in the polar vortex formation process, and could have implications for its dynamic control under nonequilibrium excitations that cause temporally intensified gradient effects (such as ultrafast laser pulses[37]).

The realistic PTO/STO superlattices are also susceptible to extrinsic influence of lattice defects, as indicated by the localized features shown in the experimental PCA (see Fig. 1 and Supplementary Fig. 2). Figure 5 illustrates the modeling of a single charge defect, which causes a compound of tilt, shift and

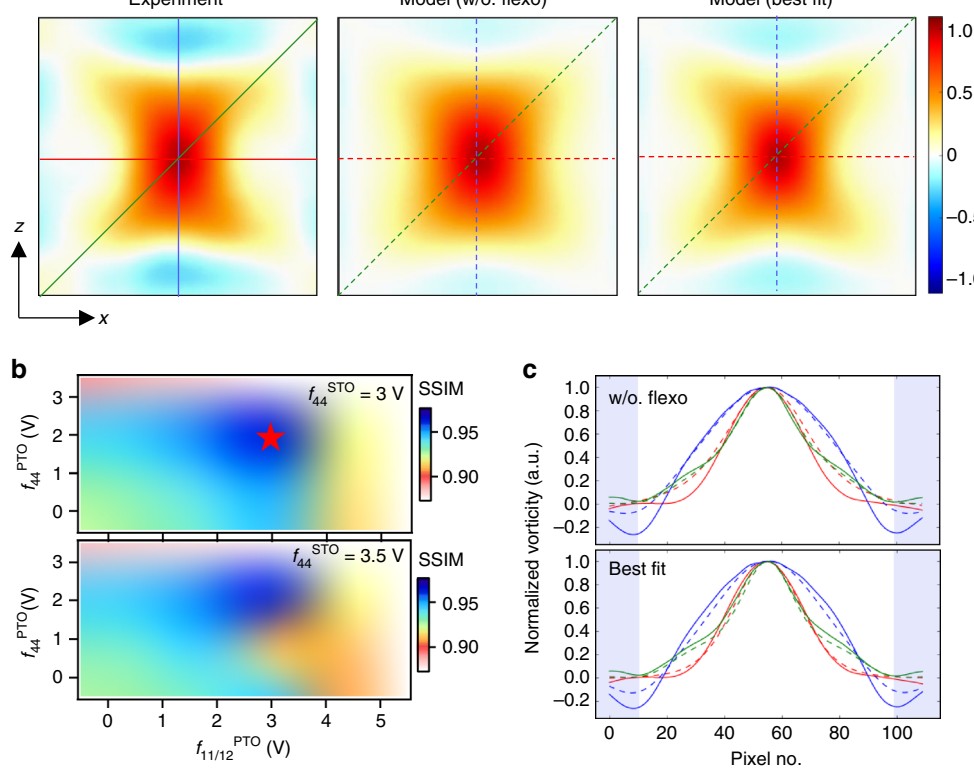

**Fig. 4** Matching the observed and simulated vortex patterns. **a** The first principal components of the observed and simulated (under zero and best-fit flexocoupling conditions) vortex patterns. The core vorticity value is normalized to unity in all cases. **b** Calculated SSIM indices for part of the simulated model library. The value of the white region is typically below 0.8. The star denotes the highest global SSIM index indicting the best-fit flexoelectric coefficients with the experimental observations. **c** Section profiles of the experimental (solid line) and simulated (dashed line) vorticity along the x, z, and xz-diagonal directions as marked and color coded in **a**. The lightly blue-shaded region corresponds to STO regions, which is excluded for the SSIM calculation. The pixel size is 45 pm

deformation to the vortex structure as a function of defect location; likewise, the defect interactions can be extracted by PCA of the simulated patterns. Note, however, that realistic defect configurations are inevitably more complex and thus require maximum experimental constraints for the modeling treatment. Further deep-learning studies (such as exemplar-based clustering[38]) are ongoing, in order to retrieve regularities among the statistically minor but still physically meaningful vortex features pertaining to defects. Our current model has neglected factors, such as the presence of finite free carriers, which could co-act with flexoelectricty on the system in a more coherent manner and thus affect the global vortex structure. Enhancing the phase-field model may potentially increase the accuracy of the determined flexoelectric coefficients.

All this detailed knowledge contributes towards a full understanding of the complex polar vortex phenomena and is a prerequisite for their utilization in emerging electronic technologies. The approach we demonstrated here may generally be applied to other mesoscopically ordered material systems, for which high-veracity atomically resolved images acquired via STEM (or scanning probe microscopy, etc.) can be harnessed for elucidating physical parameters at continuum or atomistic theory levels. Furthermore, dynamic material behavior might also be extracted from time-correlated imaging observations[39] following similar machine vision methodologies.

## Methods

**Scanning transmission electron microscopy.** STEM was performed for (PTO)$_{10}$/(STO)$_{10}$ superlattices in the high-angle annular dark-field (HAADF) mode using a spherical aberration (Cs−) corrected TEAM0.5 FEI Titan microscope operating at

300 kV. The thin-film sample fabrication details can be found in a previous study[23], where part of the STEM data analyzed in this study was also published. From the acquired HAADF-STEM images, the **PD** vectors were mapped out. In brief, the A/B-sublattice atomic positions were determined by fitting each atom scattering blob to a four-parameter, spherical Gaussian functions. Then, the displacements of each A-site cation (corresponding to one unit cell) were calculated as the difference between its position and the center-of-mass mean position of the surrounding four B-site cations. For the noise reduction purpose, the vorticity was calculated based on least-squares fitting of the $PD_x$ and $PD_z$ at each atom site with a two-variable (x and z) first-order polynomial function (i.e., a plane); 12 nearest lattice sites were included in each fit area with their **PD**'s weighted by a Gaussian function (standard deviation $\sigma \sim 0.35$ nm).

**Multivariate data analysis.** Data analysis was performed with Python 2.7 (Anaconda). Scikit-image and Scikit-learn library packages were employed for image processing, PCA, and structural similarity (SSIM) index calculations. For segmentation of the experimental vortex patterns, using an even grid of locations is unjustified due to the known imperfect global ordering conditions. Instead, the vortices were located by finding the center-of-mass of their individual patterns, disconnected after thresholding the vorticity above a certain absolute value; with the center location coordinates found, a square region with varying pixel numbers (4–6 nm in size) was then defined for each single vortex patch. For PCA, the three-dimensional (3D) (x, y, No.) dataset of all vortex square patches was reshaped and concatenated into a 2D (xy, No.) matrix N. PCA was fulfilled by singular value decomposition of the covariance matrix $C = NN^T$. The vortex feature patterns were then restored from the loading values of each obtained component. For the vortex pattern matching analysis, the simulated vorticity images were convolved with a 2D Gaussian kernel ($\sigma = 0.35$ nm, in consistence with the experimental vorticity fitting parameters) before subject to the same segmentation and PCA procedures as described above. The SSIM index is defined as:

$$\text{SSIM}(x,y) = \frac{(2\mu_x\mu_y + c_1)(2\sigma_{xy} + c_2)}{(\mu_x^2 + \mu_y^2 + c_1)(\sigma_x^2 + \sigma_y^2 + c_2)}, \qquad (1)$$

where $\mu_x, \mu_y, \sigma_x, \sigma_y$, and $\sigma_{xy}$ are the means, variances, and covariance for images x and y; $c_1$ and $c_2$ are two variables stabilizing the division with small denominator. This

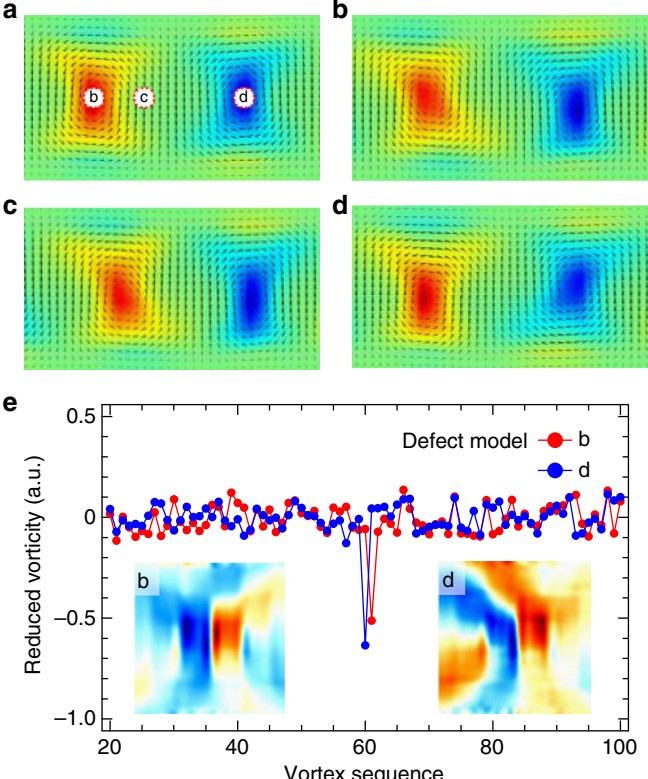

**Fig. 5** Exploring single defect–vortex interaction. **a–d** Simulated vortex pair in the models containing null **a** and a single +2q charge defect at different locations **b**–**d** as marked by dotted circles in **a**. The polarization vectors (arrows) are overlaid on the vorticity images. The flexocoupling conditions derived above were included for this modeling. **e** The eigenvectors and loading maps (inset) of the fifth PCA components of the defect models in **b** and **d**, analyzed from the full simulated vortex assemblies. In both cases, a spike in the eigenvector corresponds to the exact location of the defect and the loading map represent its main contribution to the vortex structure. Except for this and a few neighboring components that capture the perturbations to surrounding vortices (not shown), the other PCA loading map/eigenvectors are essentially identical for the two cases

SSIM index proved to show better sensitivity for our problem over other commonly used image similarity metrics, such as Euclidean distance and Pearson's correlation coefficient.

**Phase-field modeling**. In the phase-field model, the free-energy functional $F$ derived for a [001]-oriented PbTiO$_3$/SrTiO$_3$ superlattice thin film has the following form:

$$F = \alpha_{ij}P_iP_j + \alpha_{ijkl}P_iP_jP_kP_l + \alpha_{ijklmn}P_iP_jP_kP_lP_mP_n + \tfrac{1}{2}g_{ijkl}P_{i,j}P_{k,l} \\ + \tfrac{1}{2}c_{ijkl}\varepsilon_{ij}\varepsilon_{kl} - q_{ijkl}\varepsilon_{ij}P_kP_l - \tfrac{1}{2}\kappa_0E_iE_i - E_iP_i + \tfrac{1}{2}f_{ijkl}(P_{k,l}\varepsilon_{ij} - \varepsilon_{ij,l}P_k), \quad (2)$$

where all repeating subscripts imply summation over the Cartesian coordinate components $x_i$ ($i = 1, 2,$ and 3) and '$,i$' denotes the partial derivative operator with respect to $x_i$ (i.e., $\partial/\partial x_i$); $P_i$ is a component of the polarization vector **P**, $\varepsilon_{ij}$ mechanical strain and $E_i$ electric field; $\alpha$'s are the Landau expansion coefficients (up to the sixth order adopted for PTO and fourth order for STO here), $g_{ijkl}$ the polarization gradient coefficients related to the energy cost for domain wall formation, $c_{ijkl}$ the elastic stiffness tensor, $q_{ijkl}$ the electrostrictive coefficients, and $\kappa_0$ the background dielectric permittivity. The last term in Eq. (2) accounts for the energy contribution from the flexoelectric coupling with coefficients $f_{ijkl}$, which relate the strain gradient to the internal electric field and conversely, relate the polarization gradient to the elastic stress field. Another common form of flexoelectric coefficients $F_{ijkl}$ can be converted by the equations $f_{ijkl} = c_{ijmn}F_{mnkl}$. Note that the standard Voigt notation has been used in the article to simplify the flexoelectric tensor representations, for example, $f_{11} = f_{1111}$, $f_{12} = f_{1122}$, and $f_{44} = 2f_{1221}$. The material coefficients and related model parameters used are provided in Supplementary Table 1. Note that in the present model, we treated STO as simple dielectric/incipient ferroelectric layers without incorporating its structural order

parameters, given the fact that its ferroelastic phase transition temperature is well below at 300 K and thus the possible coupling strength is negligible at room temperature[40]. The mechanical strain state of the entire system was calculated with reference to paraelectric PTO and the cubic lattice constants of STO (3.905 Å), DSO (3.944 Å), and PTO (3.955 Å) were assumed.

The phase-field model was implemented in a commercial finite-element method software package (Comsol Multiphysics v.5.2) using its general partial-differential equation module. Three sets of field variables, mechanical displacement [$u, v, w$], electric potential [phi], and polarization vector $\mathbf{P} = [P_x, P_y, P_z]$, were defined in the model. From these variables, other physical quantities in Eq. (2) can be further derived, for instance, $E_x = -\text{phi}_{,x}$ and $\varepsilon_{xz} = 0.5(u_{,z} + w_{,x})$. For simplicity, a quasi-2D ($xz$-slab) superlattice model was built by adding global constraints along the $y$-direction: $P_y = 0$, $E_y = 0$, and $\varepsilon_{ij,y} = 0$ ($\varepsilon_{yy} \sim -0.3\%$ for PTO). The model consisted of 10 alternating layers of 4 nm PTO and 4 nm STO (approximately 10 unit cells for each layer thus making a total thickness of 80 nm) with a lateral dimension of 80 nm. The whole film was assumed to be constraint by a substrate (DSO) misfit strain of −0.3% and in-plane continuity periodic boundary conditions were imposed[41,42]. The displacements of the bottom surface were set to conform to the substrate, while the top surface was set free mechanically. Short-circuit electrical boundary conditions were applied for both top and bottom surfaces, and zero polarization gradients along the normal directions of these surfaces were assumed. The model geometry was discretized with a mesh of 0.8 or 0.5 nm cube elements, and a quadratic Lagrange element shape function was used as the lowest order required for calculating flexoelectricity—strain gradient is the second derivative of the displacement field. A full-3D model (i.e., no aforementioned $y$-direction constraints applied) with a smaller size of $50 \times 16 \times 50$ nm$^3$ was also tested for a few cases and showed consistent results with the quasi-2D model.

The spatio-temporal evolution of the superlattice system's field variables is governed by the time-dependent LGD polarization order parameter equation coupled with the continuum mechanical equilibrium equation and Poisson's equation of electrostatics for ideal dielectrics without the presence of free charge carriers:

$$\frac{\partial P_i(x,t)}{\partial t} = -L\frac{\delta F}{\delta P_i(x,t)}, \quad (3)$$

$$\sigma_{ij,j} = \left(\partial F/\partial \varepsilon_{ij}\right)_{,j} = 0, \quad (4)$$

$$D_{i,i} = \left(\partial F/\partial E_i\right)_{,i} = 0, \quad (5)$$

in which $t$ is the time and $L$ the kinetic coefficient related to the domain wall mobility; $\sigma_{ij}$ and $D_i$ are mechanical stress and electric displacement, respectively, as energy conjugate of $\varepsilon_{ij}$ and $E_i$. A typical vortex structure free of flexoelectricity was first obtained from an initial state consisting of small (<0.001$P_0$, $P_0 = 0.7$ C m$^{-2}$) random polarization values at 300 K. This structure was then taken as the initial state for modeling various flexocoupling conditions. In all cases, quasi-steady state solutions were calculated using the time-dependent solver after the system evolved for at least a dimensionless time length of $500(-\alpha_1/L)$.

To simulate the effects of charged point defects such as oxygen vacancies, the r. h.s. space charge term of Eq. (5) was replaced by a point source with a Gaussian distribution[43]:

$$\rho(\mathbf{r}) = \frac{Q}{\sigma^3\left(\sqrt{(2\pi)^3}\right)}\exp\left(-\frac{\mathbf{r}^2}{2\sigma^2}\right), \quad (6)$$

where $\rho(\mathbf{r})$ is the charge density in C m$^{-3}$, $Q$ is the total charge carried by defects, and $\sigma$ is the standard deviation. The attendant volumetric strain effect of defects was not yet considered. Typically, $Q = +2q$ and $\sigma = 0.3$ nm were assumed for a single defect designated in the vicinity of a vortex and in principle, multiple defects can be superimposed.

**Code availability**. The phase-field modeling codes used in this study can be made available from the corresponding authors upon request.

**Data availability**. The experimental and modeling data that support the findings of this study are available from the corresponding authors upon request.

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

## Acknowledgments

The research was performed at the Center for Nanophase Materials Sciences, a US Department of Energy, Office of Science User Facility at Oak Ridge National Laboratory. R.R. and L.W.M. acknowledge support from the Gordon and Betty Moore Foundation's EPiQS Initiative, under grant no. GBMF5307. A.R.D. acknowledges support from the Army Research Office under grant no. W911NF-14-1-0104 and the Department of Energy, Office of Basic Energy Sciences under grant no. DE-SC0012375 for synthesis and structural study of the materials. Q.L. thanks Dr. Ye Cao and Dr. Jiawang Hong for helpful discussions.

## Author contributions

Q.L. and S.V.K. conceived and designed the study. Q.L. developed Comsol phase-field models and performed the modeling and multivariate analysis of the STEM data. C.T.N and S.-L.H. performed STEM imaging and displacement vector analysis. A.R.D., A.K.Y., and M.M. fabricated the materials. L-L.L. assisted Q.L. in the STEM data analysis. S.V.K., L.W.M., and R.R. supervised the study. Q.L. and S.V.K. co-wrote the article with comments from all authors.

## Additional information

**Competing interests:** The authors declare no competing financial interests.

