## [Peer Review File · Nature Communications]

Reviewers' comments:

Reviewer #1 (Remarks to the Author):

Flexoelectricity in ferroelectric materials is appealing because of their potential ultrahigh electromechanical coupling effect. However, the characterization of the flexoelectricity strength is a challenge topic, especially in ferroelectrics due to the complex interplay between strain and domain structure. The authors proposed an interesting strategy to determine the flexoelectricity coefficients, by matching the domain structure simulated by the phase-field modeling with that derived from polar atomic displacement vectors based on STEM images. The polar vortices in thin films presented in HR-STEM images of the authors published in 2016 (Nature, 530, p198) were used to demonstrate this strategy. To filter out the statistical information of imaged domain structures and to allow a quantitative comparison of the domain structures, the domain patterns were analyzed using principle component analysis, and a quantity vorticity was applied. Though the phase-field simulation part is rather weak, the proposed strategy and the efforts of the authors for carrying out a quantitative comparison of domain structure makes the paper possible to be accepted for publication. Nevertheless, I have the following (critical) points the authors should address.

1. As mentioned in the introduction by the authors, the flexoelectricity can be modulated by and interact with different aspects, such as soft phonon modes and charge carrier transport. The domain structure images by STEM cannot exclude these influences. On the other hand, the phase-field simulation model used in the paper is rather idealized. It covers no mechanism of the above mentioned aspects. The model even excludes the (inverse) piezoelectric effect, as one can see from the free energy functional given in Eq.(2). The piezoelectric effect can largely influence the strain state and its gradient especially on the small scale and effect thus the domain structure. Since the flexoelectricity constants in this paper are fully derived from the domain structure comparison, the reliability of the results depend strongly on the validity of the phase-field model. The phase-field modeling should be thoroughly reexamined and be justified accordingly.

2. Moreover, the authors assume the flexoelectricity behavior is linear, and expected constant flexoelectricity coefficients. It has been reported in the literature (e.g. Ma, Physica Scripta T129: 180-183, 2007) that flexoelectric coefficients can be nonlinearly enhanced by strain gradient. The assumption of the linearity should be addressed, and more importantly, whether and how the proposed strategy can be adopted for the nonlinear case should be discussed. The referee assumes the strategy can run to its limit, due to the accuracy of the phase-field model.

3. What missing in the results is the information on the strain distribution, which is the basis to calculate the strain gradient and thus for the understanding of the distortion of the polar vortices. It is strongly suggested to present strain (gradient) results of the phase-field simulations and from the STEM images.

4. Regarding the phase-field modeling: 1) It is not clear to me why the mechanical behavior of the STO layer is ignored, which in fact is important to obtain a correct results on the epitaxial strain gradient across the layers. 2) The obtained domain structure is history dependent. The authors used the vortices domain structure with no flexoelectricity effect as the initial domain configuration. However, in reality the strain gradient and the flexoelectric effect should already exist at the very beginning of the formation of ferroelectric domain structure. The expected domain structure can be different. 3) In the explanation of the phase-field modeling, the boundary conditions for the polarization is not mentioned.

5. In the phase-field model, the flexoelectricity tensor f is written with four indices, but in the main body, the f is with two indices by using the Voigt notation. It should be explained clearly to it more understandable.

6. In the title "machine learning-informed phase-field modeling" is used. However, as I understand, the phase field modeling is conventional, and only the parameter study is carried out on the flexoelectricity constants. The term "machine-learning-informed" is not justified.

Reviewer #2 (Remarks to the Author):

This manuscript reports the measurement of some elements of the flexocoupling tensor in PbTiO₃ and SrTiO₃ through their influence on the (projected) spatial variation of polarization density present in an array of polar vortices in a PbTiO₃/SrTiO₃ superlattice, thereby demonstrating the role of flexoelectricity plays in the form of this polarization distribution. This is achieved by comparing atomic-resolution scanning transmission electron microscopy measurements against phase-field modelling. I leave any criticism warranted in the phase-field modeling to reviewers better qualified to comment on it, but as regards the electron microscopy and images processing, the analysis seems to me sound and compelling. The figures are of high quality and used to good explanatory effect. Though demonstrated in the context of a particular (and topical) system, I think the broader implication of the work lies in the applicability of the analysis methods to a wide range of nanoscale systems and devices where the contribution of flexoelectricity to the material's properties and behaviour is important. As such, I think this work is eminently suitable for publication in Nature Communications.

The manuscript could be published as it stands, but I encourage the authors to consider making some comment on the following points by way of minor revision to their manuscript and/or supplementary materials.

* Sublattice offset is touted as a faithful projection of a materials electric polarization. However, the differential strain that gives rise to flexoelectric effects would seem to imply that the thin film may not be flat, that there will be some variation of sample orientation across the field of view. Since the potential of local mistilt to affect sublattice shifts is well established, what impact does this have on the polarization measurements?

* The authors read a lot into the weak fluctuations visible in the polar atomic displacement vector maps: partial penetration of the ferroelectric distortion into the SrTiO₃ layers; textural features inherent to individual vortices, perhaps resulting from local structure defects. What is the case for regarding these fluctuations as significant, as opposed to being consequences of scan noise (I am aware the authors' ref. 23 gives some analysis on this point), sample texture (see previous comment), the plane-fitting approach to noise reduction and so forth? It is argued that PCA as a global statistical analysis method invariably reduces uncertainty. But, although Fig. 5 presents a compelling counter-example, interpreting meaning into individual components beyond the elbow in the scree plot is notoriously dicey, for all that their cumulative variance may be considerable.

* Perhaps too minor an issue for the present manuscript, but I am very curious as to why the structural similarity index (SSIM) outperforms a mean-square error metric here. That is not what I would expect for such smooth data.

Reviewer #3 (Remarks to the Author):

In their paper Unravelling flexoelectricity in PbTiO₃/SrTiO₃ superlattice polar vortices via machine learning-informed phase-field modeling Li et al. perform a thorough combined machine-learning – stem study to establish flexoelectricity in PbTiO₃/SrTiO₃ superlattices, and discuss possible connection to polar vortices. The study is timely and novel, and the appropriate use of machine learning is central to elucidating the effects they study. The topic and results discussed should be

of general interest and merit publication, however I have one main remark that should be addressed before considering for publication.

1. The authors clearly show, by comparing the experimental data to the simulations/machine-learning output that a specific set of flexoelectric coefficients are obtained; coefficients that are in the same vicinity as first principle results for the constituent materials. Based on the data, and the sensitivity of the polar vortex structure to flexoelectricity found from simulations, the authors claim that flexoelectricity plays an integral part in the vortex formation process. That is for me not clear from the data presented. Is it possible that the other mechanisms such as depolarization and octahedral rotations (the superlattices are grown on substrates with different symmetry (orthorhombic) compared to the constituent super-lattice materials) plays a larger role? And that due to the flexoelectric parameters for bulk PbTiO_3 and SrTiO_3 the specific vorticity is obtained once formed? The authors say in their ref. 23 that for superlattices with longer wave lengths (>18 μm) more standard flux closure is obtained with domain-walls. If there is an evolution of the vortex structure with super-lattice periodicity, does then the flexoelectric coefficients change as a function of super-lattice periodicity which one might expect if they play an important part in the vortex formation?

Reviewers' comments:

Reviewer #1 (Remarks to the Author):

Flexoelectricity in ferroelectric materials is appealing because of their potential ultrahigh electromechanical coupling effect. However, the characterization of the flexoelectricity strength is a challenge topic, especially in ferroelectrics due to the complex interplay between strain and domain structure. The authors proposed an interesting strategy to determine the flexoelectricity coefficients, by matching the domain structure simulated by the phase-field modeling with that derived from polar atomic displacement vectors based on STEM images. The polar vortices in thin films presented in HR-STEM images of the authors published in 2016 (Nature, 530, p198) were used to demonstrate this strategy. To filter out the statistical information of imaged domain structures and to allow a quantitative comparison of the domain structures, the domain patterns were analyzed using principle component analysis, and a quantity vorticity was applied. Though the phase-field simulation part is rather weak, the proposed strategy and the efforts of the authors for carrying out a quantitative comparison of domain structure makes the paper possible to be accepted for publication. Nevertheless, I have the following (critical) points the authors should address.

We thank the Reviewer for the largely positive assessment of our manuscript. We also appreciate his/her constructive comments on the phase-field modeling part. In general, we agree that the complexity of a phase-field model can always be increased by adding more free-energy terms and parameters; however, the accuracy and utility of the model may not necessarily be strengthened, if the new parameters are not available or cannot be determined from experiments. Hence, we have chosen the simplest model that captures the complexity of observed polar vortex phenomena and yet is tractable. Specifically, we start from a well-established LGD model that accounts for the polarization and polarization gradient, elastic, electrostatic and electro-mechanical coupling terms; note that our starting model is essentially consistent with that of the original work (Nature, 530, 198, 2016), only the implementation method being different. We further postulate the presence of flexoelectricity in light of recent studies on this subject (*e.g.*, refs. 9 and 22), and the introduction of this effect is found to better reproduce the ground-truth experimental observations (retrieved through PCA) with improved accuracy. Such a quantitative

match with experimental domain structures has not been achieved in previous studies and in itself, implies the adequacy of our model.

1. As mentioned in the introduction by the authors, the flexoelectricity can be modulated by and interact with different aspects, such as soft phonon modes and charge carrier transport. The domain structure images by STEM cannot exclude these influences. On the other hand, the phase-field simulation model used in the paper is rather idealized. It covers no mechanism of the above mentioned aspects. The model even excludes the (inverse) piezoelectric effect, as one can see from the free energy functional given in Eq.(2). The piezoelectric effect can largely influence the strain state and its gradient especially on the small scale and effect thus the domain structure. Since the flexoelectricity constants in this paper are fully derived from the domain structure comparison, the reliability of the results depend strongly on the validity of the phase-field model. The phase-field modeling should be thoroughly reexamined and be justified accordingly.

Response: First we would like to clarify with the Reviewer that our phase-field model does include the (inverse) piezoelectric effect. We treat this effect in an implicit manner within the fully-coupled nonlinear LGD theory framework, in contrast to some (simplified) linear models where a piezoelectric coupling term is explicitly stated—note that for proper ferroelectrics, linear piezoelectricity only holds under weak-field conditions. This can be understood by deriving the free energy functional in Eq. (2) (also see below) against the governing equations in Eq. (3):

$$F = \alpha_{ij} P_i P_j + \alpha_{ijkl} P_i P_j P_k P_l + \alpha_{ijklmn} P_i P_j P_k P_l P_m P_n + \frac{1}{2} g_{ijkl} P_{i,j} P_{k,l} + \frac{1}{2} c_{ijkl} \epsilon_{ij} \epsilon_{kl} - q_{ijkl} \epsilon_{ij} P_k P_l - \frac{1}{2} \kappa_0 E_i E_i - E_i P_i + \frac{1}{2} f_{ijkl} (P_{k,l} \epsilon_{ij} - \epsilon_{ij,l} P_k)$$

It follows, for instance, that $\sigma_{ij} = \partial F / \partial \epsilon_{ij} = c_{ijkl} \epsilon_{kl} - q_{ijkl} P_k P_l + f_{ijkl} P_{k,l}$, where the second and third rhs. terms correspond to the piezoelectric (polarization-biased electrostriction) and flexoelectric contributions to the mechanical stress. Note that here the polarization P 's are field variables whose values need to be solved self-consistently at each spatio-temporal point.

The reviewer raises concerns about other mechanisms that could influence the vortex structure, such as soft phonon modes and charge carrier transport. We do not see apparent connection between the phonon dynamics and static STEM imaging results. There could be particularly intriguing lattice dynamics associated with the vortex formation process, which presumably can be probed using ultrafast techniques. The logic is clear that our study addresses a different type of problem, in a sense, on the consequence rather than the cause. Charge carrier transport is a subtler and more relevant issue, and in principle, it can be incorporated in the phase-field model

by adding carrier density field variables governed by the drift–diffusion transport equations. The presence of high-density free carriers would modify the polarization screening and thereby interact with the domain structure. We however, believe that this is not the case in the present PTO/STO system, as with other insulating ferroelectric materials for which free charge carriers have been neglected in most theoretical studies. Some of the Authors are also involved in another (ongoing) study, where they have observed very different vortex/domain structures likely as a result of dramatically increased charge carriers.

2. Moreover, the authors assume the flexoelectricity behavior is linear, and expected constant flexoelectricity coefficients. It has been reported in the literature (e.g. Ma, *Physica Scripta* T129: 180-183, 2007) that flexoelectric coefficients can be nonlinearly enhanced by strain gradient. The assumption of the linearity should be addressed, and more importantly, whether and how the proposed strategy can be adopted for the nonlinear case should be discussed. The referee assumes the strategy can run to its limit, due to the accuracy of the phase-field model.

Response: The reviewer raised an interesting point. We agree that flexoelectricity, as with any other material parameters, can be nonlinear. However, virtually nothing has been known about this, neither from experimental nor theoretical sides. The current first-principle theories for flexoelectricity is purely linear, such as those by Hong and Vanderbilt (Ref. 3: *Phys. Rev. B* 88, 174107, 2013) and Stangel (Ref. 35: *Phys. Rev. B* 93, 245107, 2016). At the continuum level, we are not aware of any high-order free energy functionals formulated so far and not even the symmetry properties of high-order flexoelectric tensors discussed in literature. It is the Authors' opinion that high-order flexoelectricity should be considered only after a systematic and quantitative understanding of linear flexoelectricity is established (which still is a demanding task), since the introduction of the former might only contribute to the accuracy marginally.

On the other hand, the pioneering experimental studies by Ma and others were mainly based on macroscopic electromechanical measurements. This approach, as we comment in Introduction, can be highly susceptible to parasite, non-flexoelectric type of polarization (such as local electric dipoles and polar precursors), and often lead to much larger 'apparent flexoelectric coefficients'. The nonlinear polarization effects observed by these authors may not even justify intrinsic nonlinear flexoelectricity. In fact, Ma suggested in his paper (*Physica Scripta* T129, 180, 2007) that the measured nonlinear effect could be mostly due to the domain structures and thus of an extrinsic nature (apart from a pure speculation '*it is possible that a part of the nonlinearity in the*

flexoelectric effect is of intrinsic origin'). If this nonlinearity is what the Reviewer is actually interested in—if appears so since he/she commented '*the characterization of the flexoelectricity strength is a challenge topic, especially in ferroelectrics due to the complex interplay between strain and domain structure*', our study has thoroughly addressed this effect through the examinations of the vortex domain evolution under the flexocoupling.

As a brief discussion about our strategy, we agree that in principle it is possible to consider more material parameters (such as high-order ones) given accurate modeling results and reach a unique fit to the experimental output. Nonetheless, that would eventually enlarge the parametric space to an unreasonable level beyond the current phase-field modeling capabilities.

3. What missing in the results is the information on the strain distribution, which is the basis to calculate the strain gradient and thus for the understanding of the distortion of the polar vortices. It is strongly suggested to present strain (gradient) results of the phase-field simulations and from the STEM images.

Response: Following this suggestion, we have added a supplementary figure (Fig. S3) to illustrate the strain/strain gradient distributions. As a side note, the experimental strain is somewhat ill-defined due to the lack of an unstrained reference structure (as is the cubic PTO for the modeling). We have calibrated the STEM data such way that the mean in-plane lattice spacing is set equal to the nominal DSO substrate (001) d-spacing, and calculated lattice strain as the *A*- (or *B*-) sublattice spacing relative to that. The presented strain maps show correlation with the polarization gradients obtained from the same measurement, since the eigenstrain (polarization-induced strain) dominates the total strain.

4. Regarding the phase-field modeling: 1) It is not clear to me why the mechanical behavior of the STO layer is ignored, which in fact is important to obtain a correct results on the epitaxial strain gradient across the layers. 2) The obtained domain structure is history dependent. The authors used the vortices domain structure with no flexoelectricity effect as the initial domain configuration. However, in reality the strain gradient and the flexoelectric effect should already exist at the very beginning of the formation of ferroelectric domain structure. The expected domain structure can be different. 3) In the explanation of the phase-field modeling, the boundary conditions for the polarization is not mentioned.

Response: 1) The mechanical (and electromechanical coupling) behavior of STO was not ignored in our phase-field model. We have studied the flexoelectric coupling properties of the STO layers—undoubtedly, this could not be done without considering their mechanical degree of freedom. We presume the reviewer might have been misled by our statement that ‘...we treated STO as simple dielectric/incipient ferroelectric layers without incorporating its *strain order parameters*...’ (see Page 19, Paragraph 1). Here, the order parameter was meant to refer to the ferroelastic phase transition driven by the octahedral rotation (which occurs at ~ 105 K for bulk STO). We have modified it into ‘*structural order parameter*’ in the revised manuscript.

2) The Reviewer is certainly correct that the flexoelectricity acts on the system at the very beginning of domain formation. The main reason we chose an initial domain configuration formed under zero flexoelectricity is to better highlight the vortex shape/ordering modulation effects as a function of flexocoupling strength. Therefore, the same model region illustrated in Figs. 2, 3 and 5 can be compared straightforwardly. Inclusion of the flexocoupling from random states can lead to different global arrangement of polar vortices (and minor *a*-domains); however, we have verified that the vortex shape characteristics is essentially consistent for both cases as determined by the system’s free-energy. See below, for example, two sets of modeling results for the best-fit flexocoupling conditions.

Figure R1 (a) Time evolution of the free energy of the model starting from (solid line/solid marker) a pre-existing domain state and (dotted line/open marker) random states. The best-fit flexocoupling was applied in both cases.

(b) First PCA component of the two models and their comparison with the experiment results (dashed lines).

3) We have assumed zero polarization gradients along the normal directions of both top and bottom surfaces, in addition to the short-circuit electrical boundary conditions which also

contribute to the polarization evolution. The description of these boundary conditions has been added to Methods/Phase-field modeling (Page 19, Paragraph 1)

5. In the phase-field model, the flexoelectricity tensor f is written with four indices, but in the main body, the f is with two indices by using the Voigt notation. It should be explained clearly to it more understandable.

Response: The explanation about the Voigt notation has been added to Methods/Phase-field modeling (Page 18, Paragraph 1).

6. In the title “machine learning-informed phase-field modeling” is used. However, as I understand, the phase field modeling is conventional, and only the parameter study is carried out on the flexoelectricity constants. The term “machine-learning-informed” is not justified.

Response: We agree with the Reviewer that the phase-field modeling in this manuscript is conventional and largely of a parametric study type. However, the Reviewer seems to have paid less attention to the other crucial component of the manuscript, machine-learning analysis of the STEM polar-vortex data as well as the modeling output, which is highly appraised by both Reviewer #2 and #3. It is this combination of experiment and modeling that has enabled a quantitative understanding of the flexocoupling effects in the PTO/STO polar vortex system. Such a synergistic approach has never been demonstrated in previous phase-field or probably any other theory levels of modeling studies. Based on these factors, we would not regard ‘machine-learning informed’ as an over-claimed or irrelevant term, and hope to emphasize the novelty of our study by using this title.

Reviewer #2 (Remarks to the Author):

This manuscript reports the measurement of some elements of the flexocoupling tensor in PbTiO₃ and SrTiO₃ through their influence on the (projected) spatial variation of polarization density present in an array of polar vortices in a PbTiO₃/SrTiO₃ superlattice, thereby demonstrating the role of flexoelectricity plays in the form of this polarization distribution. This is achieved by comparing atomic-resolution scanning transmission electron microscopy measurements against phase-field modelling. I leave any criticism warranted in the phase-field modeling to reviewers better qualified to comment on it, but as regards the electron microscopy

and images processing, the analysis seems to me sound and compelling. The figures are of high quality and used to good explanatory effect. Though demonstrated in the context of a particular (and topical) system, I think the broader implication of the work lies in the applicability of the analysis methods to a wide range of nanoscale systems and devices where the contribution of flexoelectricity to the material's properties and behaviour is important. As such, I think this work is eminently suitable for publication in Nature Communications.

The manuscript could be published as it stands, but I encourage the authors to consider making some comment on the following points by way of minor revision to their manuscript and/or supplementary materials.

We are very pleased with and grateful for the Reviewer's highly affirmative assessment of our manuscript as well as insightful comments on the STEM data analysis.

* Sublattice offset is touted as a faithful projection of a materials electric polarization. However, the differential strain that gives rise to flexoelectric effects would seem to imply that the thin film may not be flat, that there will be some variation of sample orientation across the field of view. Since the potential of local mistilt to affect sublattice shifts is well established, what impact does this have on the polarization measurements?

Response: In this and the next comment the Reviewer raises an important issue regarding the potential for experimental artifacts and astutely highlights off-axis tilt for concern. Speaking generally regarding equating sublattice offsets to polarization we try to be explicit and utilize nomenclature (*e.g.* polar displacement) to make distinct this data represents the structural non-centrosymmetry of the cation sublattice and is not a full reconstruction of the unit cell non-centrosymmetry (as it neglects oxygen) nor a direct measure of electrical polarization (which would require insight into the effective charge distribution). It is well suited to the Z-sensitivity of HAADF STEM imaging and a powerful method of detecting polar distortions under the *a-priori* condition that they derive largely from an A-site displacive soft-mode, as is the case for PTO, but must be used cautiously otherwise. For this reason we have forgone comparison between the PTO and STO layers (as is briefly explained in Page 11, Paragraph 1).

Experimental artifacts affecting the veracity of the sublattice offsets as measures of polarization is a concern. As a sequential scan technique, scanning aberrations can lead to significant distortion from the true structure so these datasets are reconstructed from two orthogonal scans according to [C. Ophus *et al.* Ultramicroscopy 2016, 162: 1-9] to best preserve spatial information. The reconstruction does not impose a condition of uniformity, so using the STO layers as reference regions with a nominal cubic structure the deviation of the Sr and Ti sites from a uniform grid provides an upper bound for the positional error (both real deviations plus any experimental error). Heat maps of the XY offsets from uniform grid positions are shown in the figure below with σ_{rms} values of ~ 8 pm for Sr and Ti sites. Relative positional errors between Sr and Ti sublattices have an σ_{rms} value less than 5 pm.

The use of a multi-atom weighted fit to extract the spatial first derivatives of the polar displacements was necessary in the presence of even this corrected level of positional noise as it would be the dominant term using 2-pt nearest-neighbor deltas. This effectively delocalized first-derivatives and derived values such as vorticity in the range of the gaussian kernel, $\sigma = 1$ unit cell. However, the phase-field data is convolved by a matching kernel before SSIM and thus not precluded from achieving solutions with higher frequency information.

To expand on Reviewer's point on local mistilt, small off-axis tilts can manifest as sublattice offsets indistinguishable from a polar distortion in the HAADF images due to dissimilarity in channeling along different atomic columns. In this work, the presence of large area or global mistilt was screened for by considering the STO and substrate DSO as control regions, rejecting any datasets which manifested significant sublattice offsets or long-range drift thereof in these regions within the field of view. This was one of the limiting factors for including only 3 datasets

and a cumulative ~ 130 vortices in the analysis. The possibility of heterogeneous patterns of mistilt localized to the vortex domain structure, as may be implied from contribution of the shear f_{44} flexocoupling term, is conceivable as a source of experimental error dependent on scale and distribution. It is evident from the relatively good fit disregarding flexoelectric effects that texturing must be a minor effect, as might be assumed from the spatial constraints and the nominal [001] tetragonal a/c domain system.

* The authors read a lot into the weak fluctuations visible in the polar atomic displacement vector maps: partial penetration of the ferroelectric distortion into the SrTiO₃ layers; textural features inherent to individual vortices, perhaps resulting from local structure defects. What is the case for regarding these fluctuations as significant, as opposed to being consequences of scan noise (I am aware the authors' ref. 23 gives some analysis on this point), sample texture (see previous comment), the plane-fitting approach to noise reduction and so forth? It is argued that PCA as a global statistical analysis method invariably reduces uncertainty. But, although Fig. 5 presents a compelling counter-example, interpreting meaning into individual components beyond the elbow in the scree plot is notoriously dicey, for all that their cumulative variance may be considerable.

Response: We totally agree with the Reviewer's comment that we have to be rather cautious in interpreting PCA components with very small variances. And as he/she pointed out, the elbow in the scree plot can be a general rule of thumb for ascertaining statistical significance. In the manuscript, we interpret those components with $<1\%$ variance as a whole to be associated with lattice disorder effects, and Fig. 5 chiefly serves for an illustrative, rather than conclusive, purpose about the charge defect interaction and its possible retrieval through PCA. Although we are not certain after which point/component no. noise (of miscellaneous origins) becomes dominant, the association with small, widespread PCA variances would naturally agree with an incoherent and/or non-frozen disorder state in the superlattice samples. It is further proposed that deep-learning studies may reveal defect correlation information (the current vortex dataset appears not big enough for this, though), which combined with *e.g.* local spectroscopies, would provide a practical strategy to identifying defect interactions in materials. As such, we believe an exploration into the weak observed features and related modeling analysis should be beneficial to the polar vortex system and informative for studies on a wider range of nanoscale systems.

* Perhaps too minor an issue for the present manuscript, but I am very curious as to why the structural similarity index (SSIM) outperforms a mean-square error (MSE) metric here. That is not what I would expect for such smooth data.

Response: We made such a statement *a posteriori* from comparisons among several common image metrics as well as direct visual contrast. As the Reviewer may be well-versed in, SSIM has advantages over MSE and other simple metrics, including better effectiveness to small shape distortions (a more advanced version, complex wavelet/CW-SSIM would be even better). This latter attribute is pertinent for the vortex recognition task here since the shape characteristics is more important than noise/smoothness, and hence higher confidence can be expected.

Figure R2 Comparison between the structural similarity index (SSIM) and mean-square error (MSE) metrics for the vortex recognition. The inverse MSE is shown so that a high value denote a better fit in both cases.

Reviewer #3 (Remarks to the Author):

In their paper Unravelling flexoelectricity in PbTiO₃/SrTiO₃ superlattice polar vortices via machine learning-informed phase-field modeling Li et al. perform a thorough combined machine-learning – stem study to establish flexoelectricity in PbTiO₃/SrTiO₃ superlattices, and discuss possible connection to polar vortices. The study is timely and novel, and the appropriate use of machine learning is central to elucidating the effects they study. The topic and results discussed should be of general interest and merit publication, however I have one main remark that should be addressed before considering for publication.

We greatly appreciate the Reviewer’s highly affirmative assessment and would like to make a detailed discussion regarding his/her remark in what follows:

1. The authors clearly show, by comparing the experimental data to the simulations/machine-learning output that a specific set of flexoelectric coefficients are obtained; coefficients that are in the same vicinity as first principle results for the constituent materials. Based on the data, and the sensitivity of the polar vortex structure to flexoelectricity found from simulations, the authors claim that flexoelectricity plays an integral part in the vortex formation process. That is for me not clear from the data presented. Is it possible that the other mechanisms such as depolarization and octahedral rotations (the superlattices are grown on substrates with different symmetry (orthorhombic) compared to the constituent super-lattice materials) plays a larger role? And that due to the flexoelectric parameters for bulk PbTiO_3 and SrTiO_3 the specific vorticity is obtained once formed? The authors say in their ref. 23 that for superlattices with longer wave lengths (>18 uc) more standard flux closure is obtained with domain-walls. If there is an evolution of the vortex structure with super-lattice periodicity, does then the flexoelectric coefficients change as a function of super-lattice periodicity which one might expect if they play an important part in the vortex formation?

Response: From our understanding, the depolarization is a fundamental physical mechanism rather than a material property and as such, it is undoubtedly essential in the formation of polar vortices or any types of ferroelectric domain structures. We feel that flexoelectricity may not be directly comparable to the depolarization; in fact, it acts on the system partly through the modulation of depolarization fields. As concluded in the manuscript, the flexocoupling largely manifests as fine structures in the polar vortex topology. These fine structures should not be neglected and the recognition of flexoelectricity, as we envisage, may potentially provide a route to tune the structural dynamics of the polar vortices under certain experimental conditions. Note that the flexocoupling also affects the long-range vortex arrangement as briefly discussed in the article.

The Reviewer's remark on the octahedral rotation is highly pertinent. The Authors have done meticulous X-ray/electron diffraction measurements, finding no detectable signals of half-order peaks linked with octahedral rotations at (and above) room temperature. This implies an absence of the static rotation order nor even softening of the associated lattice vibration modes (since both techniques are not energy-discriminative). One possible reason is that the symmetry-mismatch from the substrates, as the Reviewer suggested, dies very quickly into the films, despite the latter being coherently strained, due to the inherent structural attributes of PTO/STO.

PbTiO₃ is known to be strongly displacive without any antiferrodistortive instabilities, while the octahedral rotation transition of SrTiO₃ occurs at a too low temperature (105 K in bulk; see *e.g.* Ref. 39: Sheng *et al*, Appl. Phys. Lett. 96, 232902, 2010) to come into play at room temperature. Therefore, we conclude that the octahedral rotation is not a crucial factor for the vortex formation in PTO/STO superlattices.

Flexoelectric coefficients are intrinsic properties of a material primarily determined by its chemical composition and crystal structure. When a material changes its form, flexoelectricity couples with the boundary conditions leading to varying structural response, as can be described by the LGD theory framework. In PTO/STO superlattices, the unit-cell periodicity regulates the vortex structure, and flexoelectricity further (simultaneously) acts on that. Our modeling shows that the vorticity of the (PTO)₁₆/(STO)₁₆ superlattice is more concentrated and the vortices are more flux-closure like with the inclusion of flexocoupling (see below). This is in qualitative agreement with the STEM observations, though currently we do not have sufficient amount of data to perform the same machine-learning analysis on the superlattices other than (PTO)₁₀/(STO)₁₀.

Figure R3 (a) HAADF-STEM image of the (PTO)₁₆/(STO)₁₆ superlattice overlaid with derived local polar displacement vectors (arrows). (b) Simulated vorticity maps of the (PTO)₁₆/(STO)₁₆ model with and without flexocoupling. (c) Vorticity profiles along the lines shown in b.

REVIEWERS' COMMENTS:

Reviewer #1 (Remarks to the Author):

Thanks for the response, but I still have two comments:

I. In the response to my point 1, the authors claimed that the second term, i.e. $-q_{ijkl} P_k P_l$ on the rhs. of their expression $\sigma_{ij} = C_{ijkl} \epsilon - q_{ijkl} P_k P_l + f_{ijkl} P_{k,l}$ is the piezoelectric contribution.

This is not correct. In fact, this quadratic term is due to the electrostriction contribution, as the electrostriction strain shows conventionally a quadratic dependency on polarization. Thereby q_{ijkl} is the electrostriction tensor.

II. In the response to my point 6, the authors confirmed that the paper deals with "machine-learning analysis of the STEM polar-vortex data" and that "phase-field modeling is conventional". Therefore, the term "machine-learning" is for STEM data and not for phase-field modeling. So the title is not justified, and it can mislead the readers.

I suggest to change the title something like

"...via machine-learning data analysis and phase-field modeling"

Reviewer #2 (Remarks to the Author):

I am satisfied with the responses the authors have provided to my queries – my recommendation for publication in Nature Communications stands. I do not feel I can usefully contribute to the discussion on the issues raised by the other reviewers.

P.S. There may be a typo in the addition defining the Voigt notation: I believe the second f_{11} should be an f_{44} .

Reviewer #3 (Remarks to the Author):

Based on the considerations of the authors as response to my questions I suggest the paper to be published.

Reviewers' comments:

Reviewer #1 (Remarks to the Author):

I. In the response to my point 1, the authors claimed that the second term, i.e. $-q_{ijkl}P_kP_l$ on the rhs. of their expression $\sigma_{ij} = c_{ijkl}\epsilon_{kl} - q_{ijkl}P_kP_l + f_{ijkl}P_{k,l}$ is the piezoelectric contribution. This is not correct. In fact, this quadratic term is due to the electrostriction contribution, as the electrostriction strain shows conventionally a quadratic dependency on polarization. Thereby q_{ijkl} is the electrostriction tensor.

The Reviewer is correct that the $-q_{ijkl}P_kP_l$ term accounts for electrostriction. However, one has to note that for most ferroelectrics (PbTiO₃ included; whose parent phase is centrosymmetric), their piezoelectricity solely originates from the electrostriction effect. We suggest two papers that could be useful to the Reviewer as well as interested readers:

[1] R. E. Newnham *et al*, *Electrostriction: Nonlinear Electromechanical Coupling in Solid Dielectrics*, J. Phys. Chem. B 101, 10141 (1997): see **Page 10144–10145, Electrostriction in Antiferroelectric and Ferroelectric Materials**.

[2] (Ref. 29 in the Main Text) Long-Qing Chen, *Phase-Field Method of Phase Transitions/Domain Structures in Ferroelectric Thin Films: A Review*, J. Am. Ceram. Soc., 91, 1835 (2008): see **Page 1836, (2) Phenomenological Description of Ferroelectric Phase Transitions**.

II. In the response to my point 6, the authors confirmed that the paper deals with “machine-learning analysis of the STEM polar-vortex data” and that “phase-field modeling is conventional”. Therefore, the term “machine-learning” is for STEM data and not for phase-field modeling. So the title is not justified, and it can mislead the readers. I suggest to change the title something like “...via machine-learning data analysis and phase-field modeling”

We appreciate the Reviewer’s discretion on this point and has modified it as ‘...via machine learning and phase-field modeling’.

Reviewer #2 (Remarks to the Author):

I am satisfied with the responses the authors have provided to my queries – my recommendation for publication in Nature Communications stands. I do not feel I can usefully contribute to the discussion on the issues raised by the other reviewers.

P.S. There may be a typo in the addition defining the Voigt notation: I believe the second f11 should be an f44.

We thank the Reviewer again for his/her recommendation. The typo has been corrected.

Reviewer #3 (Remarks to the Author):

Based on the considerations of the authors as response to my questions I suggest the paper to be published.

We thank the Reviewer again for his/her recommendation.